# Social Location and Decision-Making Among Women Living with HIV in the Southern United States: An Intersectional Approach

**DOI:** 10.3390/ijerph21121575

**Published:** 2024-11-27

**Authors:** Courtney Caiola, Marianne R. Choufani, Juliette André, Sadie B. Sommer, Alexander M. Schoemann, Sarah B. Bass, Julie Barroso

**Affiliations:** 1College of Nursing, East Carolina University, Greenville, NC 27858, USA; caiolac19@ecu.edu (C.C.); congemam17@students.ecu.edu (M.R.C.); 2School of Nursing, Vanderbilt University, Nashville, TN 62374, USA; juliette.andre@vanderbilt.edu (J.A.); sadie.b.sommer@vanderbilt.edu (S.B.S.); 3Department of Psychology, East Carolina University, Greenville, NC 27858, USA; schoemanna@ecu.edu; 4Department of Social and Behavioral Sciences, Temple University, Philadelphia, PA 19122, USA; sarah.bauerle.bass@temple.edu

**Keywords:** women, HIV, engagement in care, adherence, social location, qualitative

## Abstract

HIV care engagement and antiretroviral therapy (ART) adherence interventions aimed at decreasing viral suppression disparities for women living with HIV (WLWH) in the Southern United States (i.e., the South) are few and seldom consider diverse social locations. These refer broadly and dynamically to contextual factors and the position people occupy in a social hierarchy based on intersecting systems of oppression and social determinants of health like gender, race/ethnic, and class inequities; geographic location; and HIV-related stigma. Using an intersectional approach, we conducted in-depth interviews and used a phased approach to directed content analysis to describe women’s perceptions of their social location and how it impacts their decision-making about HIV care engagement and ART adherence. Participants were recruited to participate from a broad geographic area and represented the diverse social locations occupied by WLWH in the South. Findings from 40 in-depth interviews include descriptions of how geographic context impacts HIV care engagement and medication adherence through access to care, quality of care, and the place-based context of day-to-day experiences of living with HIV. Participants spoke of multilevel power relationships based on their gender and racial identities, and how social determinants and intersecting identities occur simultaneously and vary as a function of one another to impact health and HIV care decision-making. The findings offer a nuanced understanding of how WLWH perceive their contextually specific social locations and make critical decisions about their HIV care engagement and medication adherence.

## 1. Introduction

Viral suppression, a low or undetectable level of HIV in the body, is the cornerstone of HIV treatment and prevention because it preserves health, prevents disease progression, and effectively eliminates risk of viral transmission to sexual partners [1]. Viral suppression is achieved by successfully linking people to HIV care and supporting adherence to antiretroviral therapy (ART) [1]. Women, based on assigned sex at birth, living in the Southern region of the United States (US) have the highest rate of HIV diagnoses [2] and the lowest rate of viral suppression among women in all regions of the country [3]. Similarly, while epidemiological data monitoring transgender women are lacking, a recent multi-cohort study of transgender women in the eastern and southern US found that HIV incidence and mortality rates were highest among those residing in the South [4]. The collective high rates of HIV, low viral suppression, and high mortality among all women living with HIV (WLWH) in the South represents a stark disparity [3].

HIV care engagement and ART adherence interventions aimed at decreasing viral suppression disparities for WLWH in the South are few and have failed to consider diverse social locations [5,6,7]. These refer broadly and dynamically to contextual factors and the position people occupy in a social hierarchy based on their intersecting systems of oppression and social determinants of health like gender, race/ethnic, and class inequities; geographic location; and HIV-related stigma [6,8,9,10]. Structural and social factors such as poverty, geographic location and HIV-related stigma are driving factors in the differences by race and ethnicity [11,12,13,14]. Black/African American and Latina/Hispanic women, hereafter referred to as Black and Latina, are disproportionately poorer than other subpopulations in the US [15], a significant precipitating factor for acquiring HIV [12,13,14] and subsequent suboptimal treatment and care outcomes [14]. Moreover, since 2008, HIV diagnoses and mortality rates have consistently been the highest in the South, an area affected by high poverty rates, racial inequities, regional resource inequities, and a cultural climate that likely fosters HIV-related stigma [16]. Evidence suggests that HIV-related stigma may be intensified among WLWH [17,18] and act as a structural barrier to HIV care engagement and ART adherence [5,11]. The diverse social locations of WLWH are intersectional, meaning they are uniquely shaped by intersecting systems of oppression and competing structural and social determinants of health [9,10,19,20,21]. Few studies have explicitly examined the pathways that intersecting systems of oppression create viral suppression disparities among women living with HIV, but Norcini Pala et al. (2022) found that the negative impact is likely mediated but not fully explained by lower ART adherence and WLWH who experience more 3 or levels of severe oppression have a 66% probability of ART nonadherence compared to their peers [22].

Women’s perceptions of their social location may not always align with the sociodemographic categories typical of biomedical research approaches [23]. At the individual level, WLWH in similar sociodemographic groups (e.g., women that identify as cisgender, Black, mothers, living with HIV and in poverty) demonstrate considerable heterogeneity in how they perceive their social locations and make decisions about care engagement and ART adherence [24]. In the clinical setting, variability is often observed when people present with what appears to be very similar health conditions and demographic profiles, but based on their perceptions of their circumstances respond quite differently as they attempt to manage their health condition. Interventions are typically aimed at broad demographic groups without considering important within-group variations and how the unique needs of subgroups might vary as they respond to structural and social determinants with both vulnerability and resilience [23]. To improve HIV care engagement, support ART adherence, and decrease disparities in viral suppression rates among the diverse group of WLWH in the South, a better understanding of how WLWH perceive their social locations and make decisions about HIV care engagement and ART adherence is needed. This paper is part of a broader exploratory multistage, sequential mixed methods study [25] guided by an intersectional framework [9], which aims to develop the most salient HIV care engagement and ART adherence messaging for WLWH from diverse social locations in the South. In this paper we present the findings from initial in-depth qualitative interviews centering the voices of cis and transgender WLWH in the South to determine how WLWH perceive their social locations and make decisions about HIV care engagement and ART adherence.

### Theoretical Framework: Intersectionality

Intersectionality, the work of Black feminists and social scientists [9,10,26,27,28], offers a means of examining social location or intersecting systems of oppression and social determinants of health such as gender, race/ethnic, and class inequities; geographic location; and HIV-related stigma [9,10,29]. Intersectionality is a way of examining complexity in the human experience, including health experiences [10]. The central theoretical tenets of intersectionality include contextually specific social constructions, multilevel power relations, and simultaneity [30]. The first tenet, contextually specific social constructions, means that broad social categories like race and gender are treated as socially constructed phenomena deeply embedded in the social and contextual factors from which they are derived [30], such as geographical location. As a second tenet, intersectional scholars assert that people assigned or self-assigned to these broad social categories engage in social relationships characterized by multilevel power differentials and a social hierarchy reified by structural factors like policies, laws and rules benefiting the most powerful group(s) [30]. Use of an intersectional framework does not inherently suggest a deficit perspective; rather, it is a means of examining how people respond to different axes of power and social determinants with both vulnerability and resilience. A third core tenet posited is that social determinants and intersecting systems of oppression are not additive or multiplicative; rather, these phenomena occur simultaneously and vary as a function of one another [20]. As an example, the social location of a Black WLWH living in poverty may function quite differently than the social location of a White WLWH living in poverty; meaning, gender and class are “raced,” operating differently to influence health outcomes [20]. Simultaneity suggests that no single identity or social determinant of health comprising social location is primary, but that multiple dimensions exist simultaneously and subsequently shape health outcomes [26]. Methodologically, capturing simultaneity can be challenging, but we have applied a theoretically driven analytic approach to the qualitative data as a means for bridging the gap between this theory and praxis.

## 2. Materials and Methods

### 2.1. Sampling

A convenience sample of participants were recruited from a broad geographic area and represented many social locations occupied by WLWH in the South. Gender was self-reported and included both cis and transgender women. Race, ethnicity, class (i.e., living either above or below the federally designated poverty line), and geographic location were also self-reported. Women self-identifying as Black, Latina, and White were included as they represent the three most highly affected racial/ethnic groups of WLWH in the South. Poverty level was measured using the US Census Bureau standards [31] and geographic location was determined using the participant’s self-reported zip code as the criteria and Federal Office of Rural Health Policy designation as a rural or non-rural [32]. Specifically, this project was implemented in rural and non-rural regions of the Centers for Disease Control and Prevention’s South Census Region for HIV funding which includes the following 16 states: Alabama, Arkansas, Delaware, Florida, Georgia, Kentucky, Louisiana, Maryland, Mississippi, North Carolina, Oklahoma, South Carolina, Tennessee, Texas, Virginia, and West Virginia [33].

### 2.2. Information Power

We used the concept of information power to inform our proposed sample size [34]. There are five dimensions of information power (study aim, sample specificity, established theory, quality of dialog, and analysis strategy) and the more information the sample holds as it relates to the study purpose, the fewer participants are needed [34]. Our initial appraisal of the number of participants considered that the study purpose was somewhat broad, but that applying a well-established intersectional theoretical framework and putting mechanisms in place to foster strong interview dialog quality by having an experienced interviewer and a carefully crafted interview guide vetted by the study’s Community/Clinician Advisory Board (CCAB) enhanced the information power of the sample [34]. Additionally, we planned a theoretically driven analytic strategy that would involve both within and across case analyses [34]. Based on these considerations, a cautious provisional number (*n* = 36) was estimated and reassessed continually during the research process to ensure a comprehensive exploration of social location [34].

### 2.3. Eligibility Criteria

The eligibility criteria for participation included self-identifying as (1) a woman; (2) Black, Latina or White; (3) living with HIV or AIDS; (3) being 18 years of age or older; (4) able to read and communicate in English; and (5) mentally competent to provide informed consent.

### 2.4. Recruitment

Participants were recruited across a broad geographic area of the South, facilitated by location-specific CCAB members and study personnel at both the North Carolina and Tennessee recruitment sites, and with virtual recruitment through targeted Facebook advertising. We used a multi-pronged approach: (1) placing flyers in community-based organizations and clinics for self-referral; (2) obtaining provider referrals of WLWH both in and out of care; (3) snowball referral recruitment methods [35]; (4) distributing the self-referral flyers via social media and email listservs for HIV/AIDS support organizations to which the CCAB members and study personnel belong; and (5) targeted Facebook advertising specifically aimed at people living in rural zip codes. Eligibility and contact information were gathered on the phone, and the interviews were conducted virtually via a HIPAA-compliant virtual platform (Webex or Zoom), given recruitment began during the peak of the COVID-19 pandemic.

### 2.5. Ethical Approval

Study protocols were approved by the East Carolina University and Medical Center Institutional Review Board (#21-001403) and the Vanderbilt Institutional Review Board (#211919). A standardized virtual informed consent process was conducted with all participants who were given multiple opportunities to have any questions addressed prior to signing the digital consent form via a secure digital platform (DocuSign, Inc., San Francisco, CA, USA).

### 2.6. Study Procedures

We used qualitative descriptive methods [36,37], conducting in-depth interviews using a single interviewer with expertise in in-depth qualitative interviewing techniques. Participants were given the option of conducting the virtual interview by video or audio-only. The virtual interviews focused on the participants’ perceptions of their social locations and decision-making processes for HIV care engagement and ART adherence and experiences of stigma, discrimination, and resilience associated with HIV [36,37]. The interview questions were informed by prior research conducted by the study investigators, the HIV-related and intersectional stigma literature [6,19,38,39], and the study personnel’s collective experience working with this population. The interview questions were then vetted by the CCAB and revised accordingly prior to implementation [see Appendix A]. Self-reported demographic data were collected by study personnel using REDCap^®^ software (version 14.5.17), a secure online data management system, prior to commencing each participant interview.

### 2.7. Data Management and Analysis

A professional transcriptionist transcribed the digitally recorded interviews. The transcripts were then verified for accuracy by study personnel and imported into NVivo 12 for analysis. A core team of three study personnel (C.C., M.C., and J.A.) completed the analysis using a phased approach to directed content analysis [40] to describe women’s perceptions of their social location and how it impacts their decision-making about HIV care engagement and ART adherence. During the first phase of the analysis, each of the three study personnel familiarized themselves with the data by reading all of the transcripts [41]. They then independently coded 20% of the data (i.e., eight randomly selected transcripts) using a priori structural codes (Table 1). Not to be confused with structural issues or interventions, structural codes are a specific coding method where codes are derived from the primary theoretical concepts and research questions being explored [42,43,44]. Each of the structural codes was also assigned a valence code [43,44] denoting whether the health determinant was described by the participant as a strength (positive), a vulnerability (negative), or non-determinant (Table 1). By definition, a health determinant was considered a non-determinant when the participant indicated that it was not a factor in their health experiences or engagement and adherence outcomes. The remaining transcripts were then independently coded by two members of the analytic team. In the second phase of the analysis, the team used pattern coding [44] and examined variations in contextually specific social constructions based on geographical location. Additionally, the team analyzed the text for how participants’ social and structural valence patterns (strengths and vulnerabilities) may have influenced their reported HIV care engagement and medication adherence behaviors and for any instances in which two or more social or structural determinants co-occurred within a single statement or unit of text, indicating patterns of intersecting determinants of health (e.g., mutually constituted strengths, mutually constituted vulnerabilities, opposing determinants).

Throughout the analytic process, the analytic team members met regularly to discuss and reconcile discrepant or competing findings by adhering to the principles of study trustworthiness such as clearly linking the findings to the theoretical constructs of intersectionality and exploring rival explanations [43]. The initial coding was an iterative process, and intercoder reliability testing was carried out across several rounds using Cohen’s kappa coefficient until achieving weighted kappa values over 0.75, indicating substantial agreement [45]. The weighted range for all nodes and sources was 0.7500–0.8737 across the three coders. Calculating interrater reliability allowed the analytic team to identify codes that needed clarification, discuss areas where interpretations conflicted, discuss nuance and complexity, practice reflexivity, and refine the codebook and coding procedure [41]. We also considered field notes, practiced reflexivity by journaling personal beliefs and biases, and maintained an audit trail [44]. As a final step, the analytic team presented the preliminary findings to the research team and CCAB members to solicit feedback and refine the findings. Findings have been organized using the tenets of intersectionality to explain the central features of how the participants described and ascribed meaning to their social locations and the role that location played in their decisions regarding HIV care engagement and medication adherence.

## 3. Results

### 3.1. Participant Characteristics

Sixty-two potential participants were screened between January and September 2022. Of these, forty (*n* = 40) eligible participants completed interviews, representing seven (7) different states in the South. Data collection ceased when informational redundancy was noted [46]. The participants ranged in age from 23 to 72 years (*M* = 51.2) and reported living with HIV for 2 to 36 years with most (85%) reporting over 10 years. Most of the sample identified as Black (87.5%), living below the Federal Poverty level (57.5%), and residing in nonrural areas of the South (75%). All the participants self-identified as women (Table 2).

### 3.2. Contextually Specific Social Constructions

For the purposes of this analysis, contextual specificity was operationalized as the geographical location participants reported as their current residence based on zip code. All the participants reported residing in one of the sixteen states designated as the South Census Region for the CDC’s HIV funding and 25% of the sample reported living in a rural area. Participants described how geographic context impacts their HIV care engagement and medication adherence through access to care, quality of care, and the place-based context of day-to-day experiences of living with HIV.

#### 3.2.1. Access to Care

Most of the data from both rural and non-rural dwelling participants reflected the sentiment that WLWH who reside in rural areas face more challenges accessing HIV care and medications. Rural dwelling participants described a lack of public transportation and limited HIV care providers and facilities close to them, requiring long distance commutes to receive care. For example, one rural dwelling participant stated, “There’s no buses out here. There’s no public transportation. So, I think the nearest buses are (the) commuter bus that takes you into [city name], but you got to get to the commuter bus. No transportation out here“ [Participant HH]. Multiple transportation options and HIV care providers located within reasonable commuting distances were commonly cited by those living in suburban and urban areas. One stated, “I have more of an advantage because of my location, like I’m here in [city], and the specialty clinic is here in [city]. It’s literally like eight to 10 min away”. [Participant F]. Importantly, participants also described telehealth options and other electronic tools for care as overarching facilitators to accessing HIV care which helped them to transcend geographic location. As an example, when one rural dwelling participant was asked if their geographic location impacted their medication adherence, they responded, “No, because I’m on MyChart. MyChart is the bomb diggity, and I can request refills on there”. [Participant NN]. Interestingly, participants also made keen observations about structural determinants beyond their local contexts which negatively impact HIV care access and engagement such as when this participant stated, “*They didn’t even expand Medicaid…our state did not expand Medicaid, so that hindered a lot of people from being able to do what they do because ADAP [AIDS Drug Assistance Program] can’t do but so much for them”.* [Participant W].

#### 3.2.2. Quality of Care

Most of the data from both rural and non-rural dwelling participants reflected the sentiment that WLWH who reside in rural areas not only have difficulty accessing care but also face more challenges in receiving high quality HIV care. Participants generally attributed a higher quality of HIV care in suburban and urban contexts to more knowledgeable healthcare providers and the availability of more resources. For instance, one participant described how many of their friends had moved to more urban areas because they would be more likely to receive quality care when they offered:

Not only better care, but it’s more knowledgeable care. Because they deal with it more in those urban areas, so they know the different cases, they know what to look for instead of a doctor who’s not knowledgeable in this situation having to do their own research or send you here and there, which is nine times out of 10 going to be in a more urban area.[Participant I]

Altogether, rural dwelling participants consistently noted a lack of resources (e.g., support groups, affordable housing programs, food banks) in their specific geographic contexts as contributing negatively to their quality of health and healthcare. When one participant who had moved from a city in the Northeast to a rural area in the South was asked how living in a rural area impacted their HIV care, the participant stated, “I think it’s more of a negative than a positive. Because city life, in the city there’s way more resources. The health department that I’ve been to over here, you can’t even walk in, get a lot of condoms, you have to make an appointment”. [Participant HH].

#### 3.2.3. Place-Based Context of Day-to-Day Experiences

The geographic location of participants varied across the South (rural v. non-rural) along with day-to-day health benefits and deficits they associated with each, suggesting that placed-based context is an important determinant of health when it relates to the day-to-day experiences of WLWH. Some participants actively sought to live in more rural communities and described very positive aspects of their geographic locale such as less crime and peacefulness. Here, a rural dwelling participant described, “It’s quiet. I don’t like living close up under people. And I get that, a lot of space, I have space. And it’s quieter being outside than being in town and being outside. It just to me, living in the country is a slower pace of life” [Participant X]. Conversely, the lack of anonymity and heightened propensity to have stigmatizing experiences was embedded in the talk of both rural participants and participants who had experienced living in other regions of the country. As an example, one participant who had moved from the Northeast to a Southern city reported a heightened sense of HIV-related stigma when they regularly disclosed their status to others. The participant reported that their disclosure behaviors prompted their son, who was also living with HIV, to issue the following warning: “Mommy, down here, you don’t want to do what you do”. [Participant D]. Though participants identified both positive and negative aspects of their individual place-based contexts, a heightened experience of HIV-related stigma clearly emerged as the most impactful and co-occurring determinant.

### 3.3. Multilevel Power Relations

Multilevel power relations were operationalized in this analysis by examining the participant’s self-assigned social categories of gender and race and whether and how participants described associated power differentials acting as influencers in their experiences of HIV care engagement and adherence. Consistent with an intersectional analysis, the meaning participants ascribed to their socioeconomic class was also explored; however, the interview questions were framed in such a way as to explore the participant’s self-described “resources” and depending on the participant, may or may not have included an examination of economic resources in relation to a social hierarchy. Thus, the analytic team was not able to draw conclusions regarding the role of class in the social location of this sample of WLWH.

#### 3.3.1. Gender Identity

Most participants (*n* = 25; 62.5%) identified being a woman as a negative determinant of health, with many specifically noting subsequent negative consequences for their HIV care engagement and medication adherence behaviors. Notably, this group included both cis and transgender women. Participants reported gender inequities related to research and social service resources allocated by gender, relationship power differentials, and stigmatizing behaviors based on gender. For instance, when one participant was asked if they felt they had ever been treated differently, either positively or negatively, as a woman living with HIV, they offered:

Yes, because you hardly ever hear about a man been treated any kind of way. It’s always the woman…It’s always the woman because of…we have a cycle. When we have our menstrual, so they feel that’s…the issue is blood. So that’s why we get so much stigma and stuff, because of every month cycle that we have the blood. So, it’s like, wherever you sit, or whatever you do, they want to criticize you for, because a man don’t have that.[Participant DD]

Participants spontaneously made comparisons between how men and women living with HIV are treated. For example, one participant who attended HIV care appointments with a former male partner who was also living with HIV, contrasted the treatment their former partner received from healthcare providers by saying, “My ex didn’t get treated like that. Because he knew, like I said, but I was going to appointments with him, they didn’t do him like that”. [Participant H].

Many of the women in this group reported that such gender inequities alone or in combination with some other aspect of their identities, such as race, negatively impacted their desire to engage in HIV care or take their medication. For example, one participant noted, “…it’s always a thought that if you’re a woman living with HIV, you’re promiscuous…” [Participant O] and then went on to describe how such experiences have negatively influenced their willingness to engage in HIV care.

#### 3.3.2. Racial Identity

Most of the study sample identified as Black (*n* = 35; 87.5%), and for these participants, racial identity was evenly described as either a negative determinant of health (*n* = 16; 45.7%) or a non-determinant (*n* = 18; 51.4%), while only one Black participant (*n* = 1; 2.9%) described their racial identity as a positive determinant of health. Participants who described their racial identities as a negative determinant of health relayed experiences of racism and inequitable treatment (*n* = 16) occurring in social interpersonal interactions and within the healthcare setting. Here, a participant responded by saying, “Being Black does play a part in it because I know it’s up to me to make sure that I’m taken care of. I can’t, I don’t feel like that the way I’m treated or the doctors that I see don’t necessarily do everything that’s in my best interest” [Participant X]. This participant then denied that such experiences negatively influenced their HIV care engagement or medication adherence, but noted “…being Black, living in [state name], you have to make sure that you stand up for yourself”. [Participant X]. Black participants who described their racial identity as a non-determinant of their health or health experiences denied it influenced their experience either positively or negatively. Here, a participant responded to the questions about whether their racial identity positively or negatively impacts their experience of living with HIV when they said, “…I could see how it could impact someone, but I mean, it just hasn’t been my experience”. [Participant MM].

A smaller proportion of the women self-identified as White (*n* = 4; 10%) and only one participant identified as Latina (*n* = 1; 0.025%); thus, the analytic team was not able to draw conclusions regarding the role of their specific racial identities in the social location of this sample of WLWH. However, some consistency in the narratives of the White participants was noted across the interviews. All the White participants reported their racial identity as either a non-determinant for their health or health experiences (*n* = 3) or as a negative health determinant (*n* = 1). An interesting feature of two of the participants’ responses to whether they felt they had been treated differently based on their White identity was their inclination to pivot the narrative and discuss their negative perceptions of people of color, specifically Black people.

For example, one participant said:

I don’t think I was treated any differently, but there was comments. People that are uneducated, they’re like, “You’re White, and you got HIV? You must have been with a Black guy. Da, da, da” Stuff like that. I just ignored it, because when you are... a lot of times, when you go head to head with a Black woman about HIV, or AIDS, they seem to take it out of proportion, and I’m not saying it’s always like that, but it just seemed like there’s a comment made to me, and I’m like, “You don’t know what you’re talking about”. They seem to get out of control. But that is very seldom.[Participant CC]

The White participant who described their racial identity as a negative determinant of health cited HIV research structures as the primary source of their frustration and expressed some disenfranchisement:

Because now we have some African American studies and different study groups going on, and different focus group stuff, but they don’t want the African Americans associating with the Caucasians for HIV care for women. I’m like, “Yo, dog, I always thought I was a part of y’all, and we’ve always treated and been together, so now you trying to separate us”. I said, “What good is that?” We all living with this virus, we need to be able to be sisters and get on…[Participant BB]

### 3.4. Simultaneity

Pattern coding indicating simultaneity was noted in the data when participants spoke of social determinants and intersecting identities occurring simultaneously and varying as a function of one another. Generally, participants spoke about their health with ease and when probed about specific social determinants of health often spoke about them as intersecting in various combinations. Most of the data reflected social determinants and identified categories as working concurrently to negatively impact their health. One participant described the multidimensional nature of their social location and the negative intersection of race, gender, and HIV-related stigma when they stated:

Well, I think I’ve been treated differently for the whole realm of it all. Oh, you know? Especially being, like I said. You know, when they look at a Black woman, and the first thing comes to their mind. Okay. She’s either been out there selling her body, or she’s been doing drugs she ain’t got no business doing, and in other words all I’ve been doing is sitting home trying to take care of my children, and I happened to fall in love with a man that I didn’t know had a drug problem.[Participant CC]

The participant asserted that the specific HIV-related stigma they experience by being misperceived as a sex worker or living with a substance use disorder is mutually constituted and jointly determined by their specific racial and gender identity as a Black woman. The participant noted that such experiences had negatively impacted their HIV care engagement and medication adherence at various points in their health trajectory.

Participants described how the simultaneity of intersecting social determinants was fluid and changed based on various aspects of their identities, time, and the context of their interpersonal interactions. One participant described changes in the various aspects of their identity over time and how that had influenced their HIV care engagement and medication adherence when they offered:

I mean, there’s no one combination. Every aspect of my being, whether it’s being African American, whether it’s being transgender, whether it’s being with HIV... When I was under 25 or HIV after 25 or HIV after 30, each of them, it’s not harder, it’s just different…each aspect holds a different notch in getting care or taking medications or whatever. Any part of care, each of them have a difference to them that make it a little hard or a little easier.[Participant I]

HIV-related stigma was clearly identified as the most frequent co-occurring determinant having negative implications for women’s health, and the permutations of simultaneity, including HIV-related stigma, varied across the participant’s different social locations. This participant described how their geographical location and HIV-related stigma jointly determined their experiences of living with HIV:

I’ve dated guys and other people have told these guys, “Well, don’t mess with her. She has AIDS”. That’s one of the things that come along with being in a small town. You don’t get the opportunity to tell somebody so other people make assumptions and tell people things. Yeah, that’s the downfall of being in a small town.[Participant MM]

Participants also described instances of social determinants which intersected and worked concurrently to impact their health in positive ways. One participant offered:

…especially as the females, [we] deal with low self-esteem, and that we are not wanted, or we are ostracized. But as a female African American living with HIV, I am loving me. I feel confident in who I am. I feel encouraged and endowed to help and to share with other women that they need to get into care, to get the treatment. I am also an advocate of women that needs to know who they are, their self. Loving their self and letting them know it’s not the end of life. We’ll be okay. [Participant L]. Please see Table 3 detailing the findings and additional illustrative quotes.

## 4. Discussion

The purpose of this study was to center the voices of cis and transgender WLWH from diverse social locations in the South to determine how they perceive their social locations and make decisions about HIV care engagement and ART adherence. Women have consistently been underrepresented and understudied in HIV research [47] and in the South, the region of the country objectively most impacted by the HIV epidemic. Therefore, examining HIV care engagement and ART adherence among WLWH in the South through an intersectional lens adds to the literature by incorporating their experiences for a more comprehensive and nuanced view of the social and structural factors affecting their well-being. The study sample reflects the perspectives of a heterogeneous group of WLWH in the South and reveals the variability in their perspectives toward HIV care engagement and ART adherence. The findings from the content analysis also illuminate the jointly determined systems of gender, race/ethnic, and class inequities, geographic location, and HIV-related stigma shaping social location and specifically how WLWH in the South interpret and navigate the implications of their social locations to engage in HIV care and adhere to their medications. A major finding was how the participants’ geographic context impacted their HIV care engagement and medication adherence through access to care, quality of care, and the place-based context of day-to-day experiences of living with HIV.

The contextually specific nature of the study (the South) and its focus on health determinants of a specific condition (HIV) in a specific population (persons who identify as women living with HIV) illustrate how disease-specific phenomena like HIV-related stigma and socially constructed identities like gender may interact to create mutually constituted determinants of health [24]. At the same time, these findings demonstrate the importance of a nuanced understanding of the heterogeneity within contextually specific social locations by examining variability within seemingly similar groups of women. To compare Southern rural dwelling and non-rural dwelling persons living with HIV and simply conclude that rural-dwelling persons lag in HIV care engagement and medication adherence based on locale is not actionable, accurate, or empowering. Rurality itself is not necessarily a negative determinant of health and as demonstrated in the findings, can be positive for some who find safety and tranquility in more rural communities; rather, a more complex examination of how socially constructed identities like gender and race interact with geographical context through mechanisms like stigma is needed to create relevant and contextually specific interventions.

Our findings related to the detrimental ways unequal power relationships and simultaneity operate to shape the experiences of WLWH and their subsequent health outcomes, particularly for women of color, align with the tenets of intersectionality and offer insights into the ways WLWH navigate their social locations with both resilience and vulnerability. The way in which White participants’ narratives worked to perpetuate racial domination and reify social inequality also warrants acknowledgment and further examination of how whiteness and privilege as the dominant racial group is central to producing health inequities [26]. Additionally, given the findings highlight how particular structural inequities related to gender and race can produce poorer health outcomes, they reinforce the argument for population-based structural interventions that alter the broader context in which people make health choices [48,49].

From a clinical perspective, these exploratory findings have important implications for health practitioners as they consider the social inequities shaping their clients’ health experiences, learn to recognize the power differentials that may be present in any given clinical encounter, and examine the role they may play as providers in ameliorating or perpetuating such factors [50,51]. Clinician participation in policy advocacy is an important strategy for helping to reform the structural forces shaping the health system and HIV care. The findings also strengthen the notion that it is incumbent on healthcare providers and leaders to competently recognize that healthcare systems operate in a larger social context and address areas of structural vulnerability for the populations they serve through equitable policies and practices [49,51]. From a research perspective, the findings provide evidence for the within group heterogeneity of health experiences among WLWH and the need to continue to explore robust qualitative, quantitative and mixed method analytic approaches to capture the complexity inherent in intersectional approaches for the advancement of population health and policy formation [24].

Some limitations of this study should be noted. First, the study sample lacked diversity in certain domains such as self-identified race despite employing a multipronged recruitment strategy making some of the anticipated intercategorical intersectional analytic comparisons challenging. Second, all but two (*n* = 2) of the participants reported being at least minimally engaged in HIV care, and their experiences may not reflect the experiences of those WLWH in the South who are not at all engaged in care. Investigators may wish to consider our recruitment challenges as they design future studies and refer to the literature for reaching stigmatized populations [52,53] and executing research in a global pandemic [54,55]. Finally, the intentionally broad and open-ended interview questions related to participants’ resources did not yield a comprehensive understanding of their economic resources in relation to a social hierarchy; thus, the class component typically included in intersectional analyses was not examined in this study. However, analytical quality and trustworthiness of the presented findings were ensured through several mechanisms such as intercoder reliability [41]; regular research team meetings to discuss coding and divergent findings [43]; congruence among the study design, methods, and guiding theoretical framework [43]; creating and retaining an audit trail of all analytic procedures [43]; practicing reflexive thinking [43]; and sharing our findings with the CCAB to confirm authenticity and acceptability.

## 5. Conclusions

The diverse social locations of WLWH in the South exist at the intersection of race, gender, class, geographical location, and HIV-related stigma, yet the acknowledgment of these social determinants by the scientific community has not yielded significant progress in addressing the disparate health outcomes faced by this population of women. More nuanced healthcare practices and policies based on a clear understanding of the determinants of health experienced by WLWH in the South (e.g., intersecting racism and sexism), their relative contributions to the health outcomes of this population, and the ways in which WLWH navigate their perceived social locations with both vulnerability and strength are urgently needed as we develop the most salient HIV care engagement and ART adherence interventions for WLWH from diverse social locations in the South.

## Figures and Tables

**Table 1 ijerph-21-01575-t001:** Structural Coding Schema.

Social/Structural Determinant	Valence	Coding Symbol
Race/Racism	Vulnerability	R
	Strength	~R
	Non-determinant	nd
Gender/Gender inequity	Vulnerability	G
	Strength	~G
	Non-determinant	nd
Class/Class inequity	Vulnerability	C
	Strength	~C
	Non-determinant	nd
HIV-related Stigma	Vulnerability	S
	Strength	~S
	Non-determinant	nd
Geographical Location	Vulnerability	GL
	Strength	~GL
	Non-determinant	nd

*Notes:* R = race/racism; G = gender/gender inequity; C = class/class inequity; S = HIV-related stigma; G = geographical location; ~ denotes a strength; nd = non-determinant.

**Table 2 ijerph-21-01575-t002:** Demographic characteristics of participants (*n* = 40).

Variable	*n*	Percent
Age		
Mean	51.2	
<40	6	15
40–50	9	22.5
>50	25	62.5
Race/Ethnicity		
Black	35	87.5
White	4	10
Latina/Hispanic	1	2.5
Sex assigned at birth		
Female	38	95
Male	2	5
Gender Identity		
Woman	40	100
Man	0	0
Years Living with HIV		
<10 years	6	15
10–20 years	10	25
>20 years	24	60
Socioeconomic Status		
Above FPL	17	42.5
Below FPL	23	57.5
Geographic Location by zip		
Non-rural	30	75
Rural	10	25

**Table 3 ijerph-21-01575-t003:** Illustrative Quotes for Findings.

Finding	Quote(s)
Contextually specific social constructions (geographical location)	
Access to care	Rural dwelling participant: *So where I live, they don't have an infectious disease clinic in close proximity to where I live. So I had to travel, I don't know how many miles, but it was an hour drive.* [Participant E] Non-rural dwelling participant: *So yeah I feel like I'm easily... I'm accessible to resources and services compared to somebody who's got to drive a hundred miles to come and get help.* [Participant S]
Quality of care	Rural dwelling participant describing needing to travel for quality care: *Well, just the travel. I would say that that would be the only thing. If I had great doctors like they are and some they're interns, but if I had them here, it would be great.* [Participant NN] Non-rural dwelling participant describing her move from a rural area: *It has made a big difference, yeah. Because everything was better for me here, where I probably would've gone through more by living in a small town where there's less going on where I might not have gotten the care that I needed. Might not even be living.* [Participant B]
Place-based context of day-to-day experiences	Non-rural dwelling participant describing impact of her living environment on her day-to-day experiences: *Well, it's weird though, because before I moved up here in [city], I lived in [city, state] which is more of a urban area. I lived in the hood. I lived off of government assistance and all that. So it's actually the opposite environment that I live up here, and it's so quiet in this neighborhood and I love it here. I think I like the environment here living with HIV than I would down there in that environment.* [Participant G]
Multilevel power relations	
Gender identity	Participant describing gender discrimination as a negative determinant of health making it difficult to disclose their status and receive social support beyond their healthcare team: *But even that's something that I kind of dislike, especially as a woman, you get by society... People can tend to make assumptions and stuff, like I've definitely seen it online and whatnot, assumptions about your character or who you are based off of your status. And that's something that's made it a little bit more difficult to disclose to other people, like family or friends and stuff.* [Participant F].
Racial identity	Black participant describing race identity/racial discrimination as a negative determinant of health and how it subsequently caused her to seek care elsewhere when she was newly diagnosed: *So I've been discriminated against. I understand that, specifically because I'm black and HIV positive. I mean, it's just so out there. You do the best you can and you keep moving. I can't address it. Who am I going to tell? Who would care? And that's how most black women feel… In the beginning…I'm newly diagnosed and I get that kind of reaction. I know I'm not going to go back. You already feel bad. Because all of a sudden, now you got to deal with this, and to go into a place and then pile that on top of it, no, you don't want that. Nobody would.* [Participant H]. Black participant describing race identity/racial discrimination as a non-determinant of health: *I don't think I've been impacted by it.* [Participant A]
Simultaneity	
	Participant describing how her intersecting identities leading to oppression: *I just feel like being a woman is already of a problem of its own, and then, being African American on top of it probably triples those problems. And then, having HIV quadruples those problems because when you do come out as an African American woman with HIV, and then it's like, I have a whole lot of stigmas against me from each sector, from being African American, from being just a woman, and from having HIV.* [Participant N]

## Data Availability

The data presented in this study are available on request from the corresponding author due to privacy concerns.

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
