# Peer review of "Social Location and Decision-Making Among Women Living with HIV in the Southern United States: An Intersectional Approach"

_ijerph, 2024, doi:10.3390/ijerph21121575_

Round 1

Reviewer 1 Report

Comments and Suggestions for Authors

This paper is well-written, and the methods and results are expressed well. I am not sure that it added important new information to the literature, though it may have qualitatively confirmed results found in other studies and continues an important focus on intersectionality. I have a few suggestions for changes:

The first sentence in the abstract is a run-on sentence and could use some alterations. I suggest the following changes: "HIV care engagement and anti-retroviral therapy (ART) adherence interventions aimed at decreasing viral suppression disparities for women living with HIV (WLWH) in the Southern United States (i.e., the South) are few and seldom consider diverse social locations. These refer broadly and dynamically to contextual factors and the position people occupy in a social hierarchy based on intersecting systems of oppression and social determinants of health like gender, race/ethnic, and class inequities; geographic location; and HIV-related stigma."  This same sentence is repeated later, and I suggest the same changes to be made.

Table 2 has no explanation for the definition of rural and non-rural, and this should be explained in the body of the paper

Reviewer 2 Report

Comments and Suggestions for Authors

The article, Social Location and Decision-Making Among Women Living with HIV in the Southern United States: An Intersectional Approach, provides valuable insights into how intersecting social identities—such as race, socioeconomic status, and gender—impact health-related decision-making among women living with HIV in the Southern U.S. The study’s objectives are well-stated and address a critical research gap, focusing on the compounded social challenges faced by these women in a region marked by high levels of poverty, limited healthcare access, and enduring health disparities. The intersectional framework employed is appropriate and provides a nuanced perspective on how social determinants influence health choices for this demographic.

Some minor improvements could enhance the clarity and impact of this work. The introduction could benefit from a slightly expanded literature review, offering a broader context of studies addressing intersectionality and health disparities, especially within the Southern U.S. This would reinforce the study’s importance and more clearly position it within existing research. The authors might also benefit from a more explicit definition of intersectionality in the introduction, to reinforce its application throughout the analysis.

The methodology is generally sound, employing a qualitative approach that allows for in-depth exploration of participants’ experiences. The study would be further strengthened by providing a bit more detail on sample selection and data collection methods, as well as an overview of data analysis techniques to enhance methodological transparency. While the authors mention coding themes from interviews, additional detail on how intersectionality was applied within this coding process would give readers a better understanding of the analytical depth.

The findings are well-organized and presented through relevant themes that reveal the significant barriers to decision-making for these women. Quotations from participants add richness to the results; however, the inclusion of a few more contextually interpreted quotes could provide even clearer insights into how specific social identities impact their health-related choices. The discussion ties the findings back to the intersectional framework effectively, and the authors’ interpretations are well-supported by the data.

The conclusion offers meaningful insights and underscores the need for targeted policy and healthcare interventions. To strengthen this section, the authors could consider including more specific recommendations, such as advocating for culturally tailored healthcare practices and policies that address economic and racial barriers unique to the Southern U.S. Overall, the article makes an important contribution to public health and intersectional studies, and with these minor revisions, it will provide even clearer guidance for future research and practice.

Please remove self-citations.

Reviewer 3 Report

Comments and Suggestions for Authors

This work presents the findings from WLWH in the Southern US to determine how they perceive their social locations and make decisions about HIV care engagement and ART adherence through a qualitative interview.

Please give clarity on coding symbols used in table 1

Table 2-Its better to denote the second category under the section ‘Years living with HIV’ as 10-20 rather than >10

I suggest to have a consolidation table to communicate the themes derived (and the findings) such as access to care, quality of care, Place-based context of day-to-day experiences, Multilevel Power Relations, Gender identity, Racial identity and Simultaneity. This will enhance readability.

Discussion section covers the implication of major findings of the study
